# Bcl-2 Up-Regulation Mediates Taxane Resistance Downstream of APC Loss

**DOI:** 10.3390/ijms25126745

**Published:** 2024-06-19

**Authors:** Angelique R. Wise, Sara Maloney, Adam Hering, Sarah Zabala, Grace E. Richmond, Monica K. VanKlompenberg, Murlidharan T. Nair, Jenifer R. Prosperi

**Affiliations:** 1Department of Biochemistry and Molecular Biology, Indiana University School of Medicine South Bend, South Bend, IN 46617, USA; awise@nd.edu (A.R.W.); saramaloney7196@gmail.com (S.M.); pro.ahering@gmail.com (A.H.); sbzabala914@gmail.com (S.Z.); monica.vanklompenberg@gmail.com (M.K.V.); 2Harper Cancer Research Institute, South Bend, IN 46617, USA; grichmon@nd.edu; 3Department of Biological Sciences, University of Notre Dame, Notre Dame, IN 46556, USA; 4Department of Biology, Indiana University—South Bend, South Bend, IN 46634, USA; mnair@iusb.edu; 5Department of Computer Science and Informatics, Indiana University—South Bend, South Bend, IN 46634, USA

**Keywords:** breast cancer, APC, Bcl-2, chemoresistance, paclitaxel

## Abstract

Triple-negative breast cancer (TNBC) patients are treated with traditional chemotherapy, such as the taxane class of drugs. One such drug, paclitaxel (PTX), can be effective in treating TNBC; however, many tumors will develop drug resistance, which can lead to recurrence. In order to improve patient outcomes and survival, there lies a critical need to understand the mechanism behind drug resistance. Our lab made the novel observation that decreased expression of the Adenomatous Polyposis Coli (APC) tumor suppressor using shRNA caused PTX resistance in the human TNBC cell line MDA-MB-157. In cells lacking APC, induction of apoptosis by PTX was decreased, which was measured through cleaved caspase 3 and annexin/PI staining. The current study demonstrates that CRISPR-mediated APC knockout in two other TNBC lines, MDA-MB-231 and SUM159, leads to PTX resistance. In addition, the cellular consequences and molecular mechanisms behind APC-mediated PTX response have been investigated through analysis of the BCL-2 family of proteins. We found a significant increase in the tumor-initiating cell population and increased expression of the pro-survival family member Bcl-2, which is widely known for its oncogenic behavior. ABT-199 (Venetoclax), is a BH3 mimetic that specifically targets Bcl-2. ABT-199 has been used as a single or combination therapy in multiple hematologic malignancies and has shown promise in multiple subtypes of breast cancer. To address the hypothesis that APC-induced Bcl-2 increase is responsible for PTX resistance, we combined treatment of PTX and ABT-199. This combination treatment of CRISPR-mediated APC knockout MDA-MB-231 cells resulted in alterations in apoptosis, suggesting that Bcl-2 inhibition restores PTX sensitivity in APC knockout breast cancer cells. Our studies are the first to show that Bcl-2 functional inhibition restores PTX sensitivity in APC mutant breast cancer cells. These studies are critical to advance better treatment regimens in patients with TNBC.

## 1. Introduction

Triple-negative breast cancer (TNBC) is clinically challenging because it lacks the estrogen receptor (ER), progesterone receptor (PR), and amplification of HER2. Consequently, patients with this disease cannot be treated with therapy directed at ER and HER2 and must rely solely on chemotherapy as the standard of care [1,2]. Approximately 40–50% of women with TNBC treated with chemotherapy achieve a complete pathological response [1,2]. The remaining patients often develop drug resistance, tumor recurrence, and increased metastatic spread, making this a major driver of breast cancer mortality. Together, these factors result in TNBC patients having the lowest five-year survival rate compared to the other subtypes, underscoring the urgent need for new therapeutic options for these patients.

One attractive candidate in TNBC is the tumor suppressor Adenomatous Polyposis Coli (APC). APC expression is lost in up to 70% of sporadic (non-hereditary) breast cancers, including TNBC. We and others have implicated APC in the control of several chemoresistance pathways. Located on the long arm of chromosome 5, APC is primarily recognized for its pivotal role in colorectal cancer (CRC), where truncating mutations often occur in the mutation cluster region (MCR). APC loss through mutation or promoter hypermethylation occurs in multiple cancer types, including colon, prostate, breast, and non-small cell lung cancers [3,4,5,6,7,8,9,10]. APC loss can also occur through upregulation of miR-135 in the development of colorectal, breast, and gastric cancer [11,12,13,14,15], making APC a viable therapeutic target in these cancer types. In addition, APC loss correlates with decreased overall survival in non-small cell lung cancer (NSCLC) and breast cancer [3,16]. These previous studies are critical in demonstrating that APC loss could serve as a therapeutic marker and address the question of why patients with APC-deficient tumors have a worse prognosis than patients with APC-competent tumors. To establish the clinical relevance of APC loss in breast cancer, we previously showed that APC mRNA and protein expression is decreased in primary human breast tumors compared to normal tissue [17,18,19]. Understanding the contribution APC plays in mediating chemoresistance can help enhance the therapeutic guidelines for cancer types with chemotherapy as the only option, such as TNBC.

TNBC is typically treated with a chemotherapy regimen that includes anthracyclines and taxanes. We have previously demonstrated the role of APC loss in regulating response to the anthracycline doxorubicin (DOX) through modulation of the DNA damage repair pathway and drug export pumps, such as MDR1. In the current study, we have investigated how loss of APC results in resistance to the taxane drug paclitaxel (PTX). In a model of metaplastic breast cancer, MDA-MB-157 cells, we showed alterations in the cell cycle regulators Cyclin B and CDK1, suggesting uncontrolled proliferation even in the presence of PTX treatment. Here we have used the gold standard cell model of TNBC, MDA-MB-231 cells, in addition to the SUM159 cells, to demonstrate that regulators of apoptosis and tumor-initiating cells (TICs) are altered with loss of APC and contribute to the lack of response to chemotherapy.

## 2. Results

### 2.1. Loss of APC Leads to PTX Resistance

Our previous studies demonstrated that APC knockdown in MDA-MB-157 cells resulted in resistance to PTX [20]. Therefore, we first wanted to validate that loss of APC broadly causes PTX resistance in TNBC. Using the CRISPR/Cas9 system in two TNBC cell lines, MDA-MB-231 and SUM159, we generated APC knockout cell lines. Starting with pooled cell lines from four different gRNAs, we used two gRNAs to generate clones of APC knockout cells and a single clone of a non-targeting control (NTC). APC expression was significantly decreased in the APC knockout clones compared to the parental and NTC control cells in both the MDA-MB-231 (Figure 1A) and SUM159 (Figure 1B) cell lines.

We next investigated how APC loss would impact response to PTX. Using the IC50 concentration of PTX in the MDA-MB-231 or SUM-159 cells (85 nM and 42.7 nM, respectively), we assessed the apoptotic response using annexin V/PI staining and flow cytometry. In both MDA-MB-231 and SUM-159, PTX induced a robust increase in apoptosis in the NTC cells; however, APC clone 1 and APC clone 2 cells were resistant to the apoptosis induction by PTX (Figure 1C,D). Taken together, these data demonstrate that loss of APC in the human TNBC cell lines MDA-MB-231 and SUM-159 results in resistance to PTX treatment, which is similar to our previous findings in the metaplastic MDA-MB-157 cells [20]. Due to the robustness of the MDA-MB-231 cells in the TNBC literature, the remaining studies were performed in this line. 

### 2.2. In Vitro Tumorigenic Assays Show Alterations in Clonogenic Growth and ALDH+ Cell Population

To investigate the tumorigenic potential of the MDA-MB-231 cells with loss of APC, in vitro tumorigenic assays were performed. Surprisingly, APC loss had no impact on cell proliferation or migration in a standard scratch assay (Figure 2A,B). Previous investigations have shown that the tumor-initiating cell (TIC) population is enriched in a chemoresistant population of breast cancer [21]. Therefore, we next assessed the TIC population using both a clonogenic assay and an Aldefluor assay. The APC^KO^ cells showed an increase in individual colony area (Figure 2C) and exhibited an increase in Aldefluor activity (Figure 2D), which suggests that the TIC population is increased by loss of APC in the MDA-MB-231 cells. 

### 2.3. Alterations in Cell Cycle and Apoptosis Proteins

We previously demonstrated that APC knockdown in MDA-MB-157 cells resulted in increased expression of the cell cycle regulator CDK1 [22]. Therefore, we investigated the expression of CDK1 in the APC^KO^ MDA-MB-231 cells and found that only the clone 1 cells showed an increase in CDK1 expression (Figure 3A), suggesting that combination treatment with a CDK1 inhibitor may not overcome the PTX resistance. This led us to explore other molecular markers downstream of APC loss.

In addition to alterations in cell cycle proteins, RNA-sequencing data using the MDA-MB-157 cells with APC knockdown demonstrated that transcripts involved in apoptosis were altered [22] (Appendix A). Using the MDA-MB-157 cells, we then assessed the expression of a panel of BCL-2 family proteins and found an increase in the namesake and pro-survival protein Bcl-2 (Appendix A). Similarly, we found that there is also a significant increase in Bcl-2 in the CRISPR-mediated APC^KO^ MDA-MB-231 cells (Figure 3B).

### 2.4. Blocking Bcl-2 Function Enhances Response to PTX in Cells Lacking APC Expression

Given that expression of the pro-survival protein Bcl-2 is increased in both clones lacking APC expression, we sought to understand whether blocking the function of Bcl-2 would re-sensitize cells to PTX. To facilitate the blockade of Bcl-2, we turned to the BH3 mimetic, ABT-199 (Venetoclax). We chose to investigate ABT-199 because of its selectivity for Bcl-2 only, as this drug does not inhibit the other pro-survival members of the BCL-2 family (such as Bcl-xL or Mcl-1). Using the MDA-MB-231 TNBC cells (NTC or clones), we treated the cells with either PTX, ABT-199, or a combination of both for 24 h. Apoptosis was measured by Annexin/PI using flow cytometry. As shown earlier, treatment of the NTC cells with PTX induces a robust apoptotic response, which is not enhanced by the addition of ABT-199 (Figure 4). However, the APC clonal cells show no apoptosis induction by either PTX or ABT-199 alone, while the combination induces a significant apoptotic response (Figure 4). These data demonstrate that targeting Bcl-2, using BH3 mimetics, sensitizes APC^KO^ cells to PTX treatment in vitro. 

## 3. Discussion

Paclitaxel (PTX) is a taxane chemotherapeutic agent, which functions by disrupting the microtubule assembly process. While it has a well-characterized mechanism of action, less is known about the development of resistance, specifically in TNBC. Furthermore, few studies have demonstrated the connection between loss of the APC tumor suppressor and resistance to chemotherapy. Here we have demonstrated that loss of the APC tumor suppressor using CRISPR-based technology in two TNBC lines (SUM159 and the gold standard model of MDA-MB-231 cells) results in resistance to PTX. While we had previously shown PTX resistance as a response to APC loss, those studies were performed in the metaplastic breast cancer cell line MDA-MB-157 [20,22]. The current study suggests that this is a widespread impact of APC loss in TNBC regulating chemoresistance to PTX. 

In addition to the response to therapy, we investigated the classical in vitro markers of tumorigenic potential. Surprisingly, there was no change in cell proliferation with loss of APC. However, this correlates with previous work from our laboratory showing that murine mammary tumor cells have no change in proliferation based on APC status [23]. This may be a result of the lack of Wnt pathway activation seen in these models [23], or it may indicate a difference in the function of APC in the mammary gland as opposed to other tissue types. In addition, no change was observed in the ability of cells to close a wound, indicating no alteration in migratory capacity. 

Tumor-initiating cells (TICs) are known to be resistant to chemotherapy, due to their slower growing nature [24,25]. Our previous studies have also shown a correlation between APC status and TICs [17,20,26]. Therefore, we used two methods to investigate the TIC population in the MDA-MB-231 cells. The clonogenic assay and Aldefluor assay demonstrated that APC loss resulted in increased individual colony area and increased fluorescence, respectively. Combined, these findings suggest that loss of APC increases the TIC population, which could result in resistance to PTX.

Given that there are multiple gene expression changes related to apoptosis signaling present in the APC shRNA cell line compared to the MDA-MB-157 cells (Appendix A), coupled with the increase in Bcl-2 expression, we sought to investigate the intrinsic pathway of apoptosis in this model. To do this, we turned our attention to the B cell lymphoma 2 (BCL-2) family, which is comprised of both apoptosis stimulating and inhibiting proteins that function as regulator and effector proteins. Overexpression of the pro-survival members, Mcl-1, Bcl-XL, and Bcl-2, is related to disease prognosis and response to chemotherapy [27]. While our current data and previous publications have shown no interaction between APC and Mcl-1 [26], the prior literature suggests that Mcl-1 stability is crucial to halt the progression of the cell death pathway. This discrepancy may be dependent on the tissue type investigated. The anti-apoptosis proteins, Bcl-XL and Bcl-2, bind to pro-apoptotic proteins, Bax or Bak. The sequestration of this heterodimer directly inhibits the permeability of the mitochondrial membrane, the release of cytochrome c, and the continuation of the apoptotic cascade. The ratio of Bcl-XL/Bcl-2 to Bax is tightly regulated, with an imbalance in expression leading to suppressed apoptosis. Furthermore, overexpression of the anti-apoptosis proteins has been associated with chemoresistance [27]. Genetic variants of Bcl-2 have been connected to increased resistance to PTX [28], which clearly points to the involvement of Bcl-2 in mediating resistance to PTX-induced apoptosis. Previous clinical trials have shown mixed results of ABT-199 in combination therapy for ER+ or Her2+ breast cancer [29,30]. In the current study, we have demonstrated that the Bcl-2 specific BH3 mimetic, ABT-199, restores sensitivity to PTX in APC^KO^ TNBC cell lines.

## 4. Materials and Methods

### 4.1. Cell Culture

MDA-MB-231 TNBC cells (ATCC, Manassas, VA, USA) were maintained at 37 °C with 5% CO_2_ in DMEM media with 1:5000 plasmocin, 1% penicillin/streptomycin, and 10% fetal bovine serum. SUM159 TNBC cells (a gift from Z. Schafer’s lab at Notre Dame) were maintained at 37 °C with 5% CO_2_ in Ham’s F12 media with 1:5000 plasmocin, 1% penicillin/streptomycin, and 10% fetal bovine serum. Cells were regularly passaged using 0.25% trypsin/EDTA and were authenticated using STR DNA profiling (Genetica DNA Laboratories, Burlington, NC, USA). For drug treatments, MDA-MB-231 cells were treated at 50–70% confluence with 85 nM paclitaxel (Sigma, St. Louis, MO, USA), 2.25 uM ABT-199 (Cayman Chemical, Ann Arbor, MI, USA), or control DMSO for 24 h. SUM159 cells were treated with 42.7 nM paclitaxel. 

For cell proliferation assays, MDA-MB-231 NTC and APC^KO^ cells were plated at a density of 1 × 10^4^ cells per well in a 24-well plate. Cells were counted daily from days 3 through 7 post-plating. 

### 4.2. CRISPR-Mediated Knockout

We have used the TNBC cell lines, MDA-MB-231 and SUM159, to generate APC knockout cells using CRISPR/Cas9. Four guide RNA sequences (gRNA) and one non-targeting control gRNA were selected from the human GeCKO library [31] and cloned individually into Lenti-CRISPR v2. Plasmids were prepped using miniprep (QIAprep Spin Miniprep kit, Qiagen, Germantown, MD, USA) and maxiprep (GenElute HP Plasmid Prep kit, Sigma, St. Louis, MO, USA), then added to human embryonic kidney HEK293FT cells with packaging plasmid (psPAX2) and envelope plasmid (pMD2.G) in Lipofectamine 3000 (Invitrogen, Carlsbad, CA, USA) and Opti-MEM (Gibco, Grand Island, NY, USA). Fresh media was added to the cells 24 h post-transfection. The lentiviral supernatant was removed 48 h after replacing the media, centrifuged, and filtered through a 0.45 uM PES filter. MDA-MB-231 or SUM159 cells were plated at a seeding density of 30,000 cells/well to be roughly 75% confluent at the time of transduction. Cell media was removed, and lentiviral titer was added to two 6-well plates of MDA-MB-231 or SUM159 cells. Lentivirus was left on for 24 h and then replaced with normal media. Forty-eight hours post media replacement, media with puromycin was added to select for successfully transduced cells. Cells were then plated at 1000 cells/well in a 100 mm plate to isolate single-cell colonies from pools of cells.

### 4.3. Protein Isolation and Western Blot Analysis

Whole cell lysates were isolated using lysis buffer (20 mM Tris-HCl, 150 mM NaCl, 1% Triton-X, 0.5% NP-40, 50 mM NaF, 1 mM Na_3_VO_4_, 5 mM sodium pyrophosphate, 0.2 mM PMSF, 1× protease inhibitor cocktail (Fisher, Waltham, MA, USA), and 1× phosphatase inhibitor cocktail 2 (Sigma, St. Louis, MO, USA)), and a BCA assay (Thermo, Waltham, MA, USA) was performed to determine protein concentration. Samples (protein and sample buffer) were boiled for 5 min prior to gel loading. Prepared samples of 30 ug of protein were separated by SDS-PAGE gel and transferred onto Immobilon P membrane (Millipore, St. Louis, MO, USA). Membranes were blocked in 5% NFDM for 1 h at room temperature. Blots were probed with the following primary antibodies overnight at 4 °C: APC, Bcl-2, Bad, Bcl-xL, Bim, Bak, Bax, Mcl-1, Bid, cleaved caspase 3 (1:1000), and actin (1:25,000). After secondary antibody incubation, blots were developed using either Clarity or Max Clarity regent on a ChemiDoc MP Imaging System (Bio-Rad, San Francisco, CA, USA). Analysis was performed using ImageJ software (Version 1.35K, NIH) to measure densitometry.

### 4.4. Clonogenic Assay

MDA-MB-231 NTC and APC^KO^ cells were plated in duplicate at a density of 500 cells per well in a 6-well plate. The media was replaced every 3–4 days, and colonies were allowed to form for 8 days, at which point, they were fixed with 3.7% formaldehyde. After washing in water, colonies were stained using 0.01% crystal violet for 45 min. Images of each well were taken from a fixed height with consistent white background lighting. Analysis was performed using ImageJ (Version 1.35K, NIH) and particle analysis. 

### 4.5. Migration Assay

MDA-MB-231 NTC and APC^KO^ cells were plated in duplicate at a density of 1 × 10^5^ cells per well in a 6-well plate. Once cells reached confluence, four scratch wounds in the shape of a number sign (#) were made in each well with a 200 μL pipette tip. The media was changed, a zero-hour time point image was acquired, and 2.5 μg/mL mitomycin C (Sigma) was added to inhibit proliferation. Phase-contrast images were acquired at 0, 24, and 48 h post-scratching using an EVOS inverted microscope with a 20× objective and Sony ICX285AL CCD camera. NIH ImageJ (Version 1.35K) was used for analysis. 

### 4.6. Statistical Analysis 

All values are reported as mean +/− SD. Data from treatment groups were compared using one-way ANOVA with a post hoc Tukey’s *t*-test. All graphs were generated, and statistics were performed using GraphPad Prism version 10.1.1. for MacOS, GraphPad Software (La Jolla, CA, USA). 

## Figures and Tables

**Figure 1 ijms-25-06745-f001:**
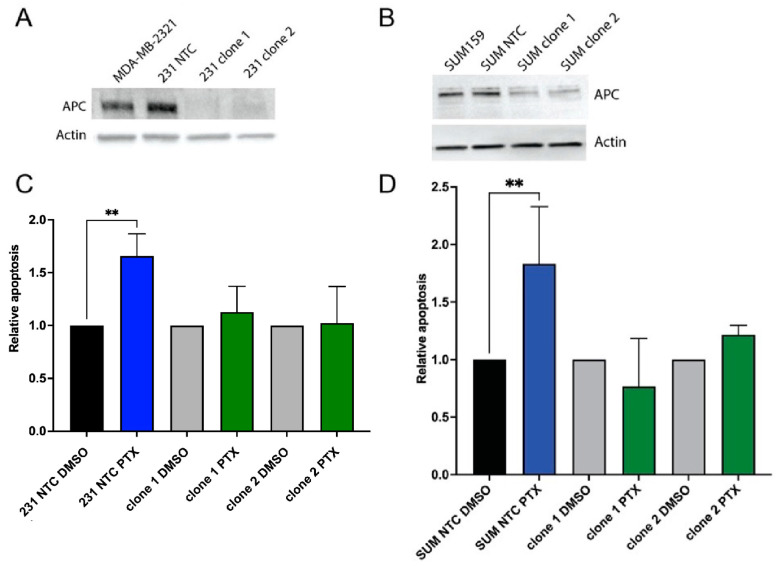
APC status in TNBC cell lines impacts therapeutic response. CRISPR/Cas9 knockout of APC in (**A**) MDA-MB-231 or (**B**) SUM159 cells. Protein from non-targeting control (NTC) and two gRNA-mediated clonal cell lines was run on an SDS-PAGE gel and probed for APC and actin. Blots are representative (n = 3). (**C**,**D**) Treatment with PTX induces apoptosis (annexin V/PI staining) in NTC cells. However, clonal APC knockout cells from both MDA-MB-231 (**C**) and SUM159 (**D**) show no induction of apoptosis. The graphs show relative PTX-induced apoptosis, compared to DMSO-treated cells. Experiments were performed 3 independent times, and a one-way ANOVA was used to determine significance (** *p* < 0.01).

**Figure 2 ijms-25-06745-f002:**
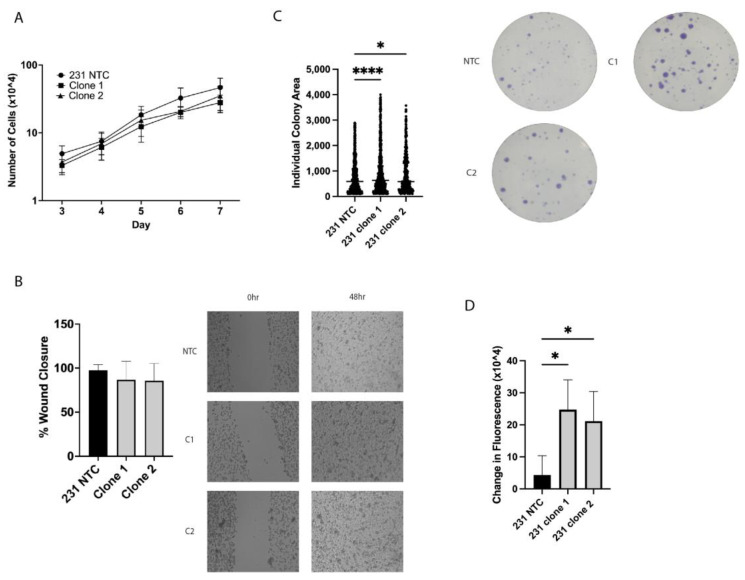
APC loss alters in vitro tumorigenic phenotypes. (**A**) Cell counting assay of MDA-MB-231 NTC and APC^KO^ cells. Cells were plated and counted from day 3 through day 7 and showed no difference in growth. (**B**) Wound healing assay showed no change in the ability of cells to fill a scratch over 48 h. Representative images show the original scratch (0 h) and the filled wound (48 h). Images were taken with an EVOS inverted microscope with a 20× objective and Sony ICX285AL CCD camera (**C**) Clonogenic assay demonstrated increased individual colony area in the 231 clone 1 and clone 2 compared to the 231 NTC cells. Representative images taken with a fixed height camera and a light box show the stained colonies after 8 days in culture. (**D**) Overall change in fluorescence between control and test samples in an Aldefluor assay showed increased ALDH activity in the 231 clone 1 and clone 2 compared to the 231 NTC cells. Experiments were performed 3 independent times, and a one-way ANOVA was used to determine significance (* *p* < 0.05; **** *p* < 0.001).

**Figure 3 ijms-25-06745-f003:**
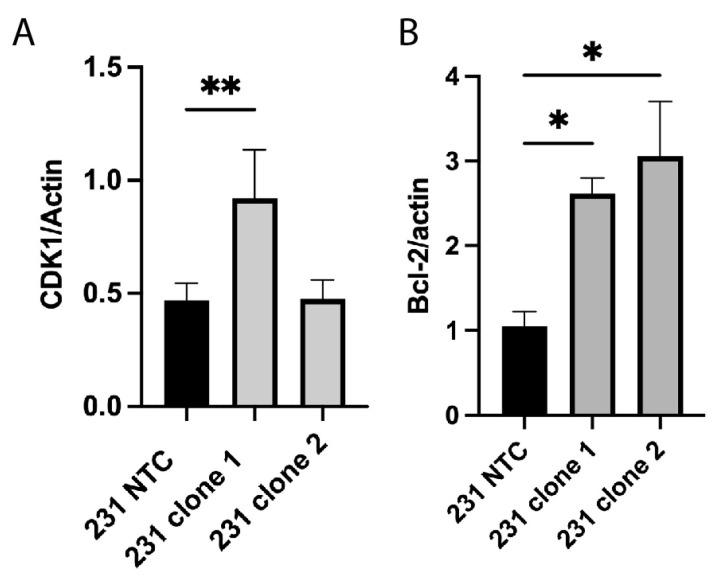
Expression changes in proteins involved in cell cycle and apoptosis. (**A**) CDK1 expression is increased by western blot in the 231 clone 1 cells but not the 231 clone 2 cells compared to the 231 NTC cells. (**B**) Bcl-2 expression is increased in both MDA-MB-231 APC knockout clones compared to control. Experiments were performed 3 independent times, and a one-way ANOVA was used to determine significance (* *p* < 0.05; ** *p* < 0.01).

**Figure 4 ijms-25-06745-f004:**
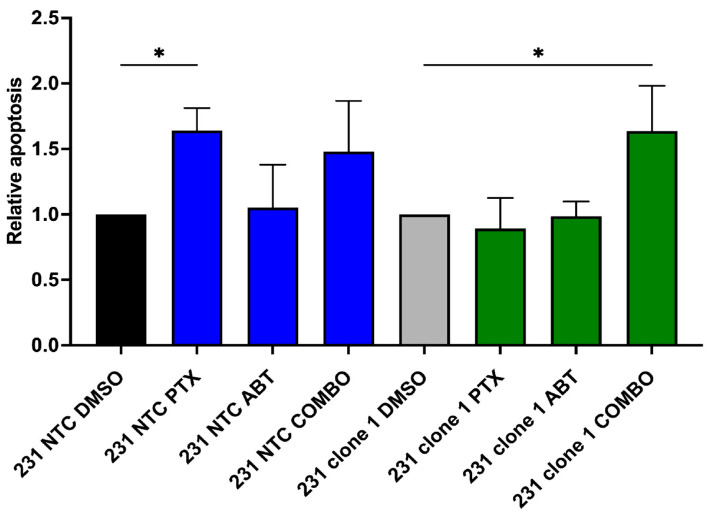
Reversal of Resistance with ABT-199 Combination Treatment. The IC50 for the Bcl-2 specific BH3-mimetic (ABT-199) in MDA-MB-231 cells was determined (2.25 uM). This concentration was used alone or in combination with PTX treatment. APC control (231 NTC) or knockout cells (231 clone 1) were treated with ABT-199 and/or PTX for 24 h, and apoptosis was measured through annexin V/PI staining. After flow cytometric analysis, we observed that while the clones are resistant to PTX-induced apoptosis, the combination treatment induced a robust apoptotic response (n = 3) * *p* < 0.05 with one-way ANOVA.

## Data Availability

The data presented in this study are available on request from the corresponding author. The data are not publicly available due to publication in preparation.

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
