# Peer review of "Bcl-2 Up-Regulation Mediates Taxane Resistance Downstream of APC Loss"

_ijms, 2024, doi:10.3390/ijms25126745_

Round 1

Reviewer 1 Report

Comments and Suggestions for Authors

I read with great interest the article "Bcl-2 up-regulation mediates taxane resistance downstream of APC loss" by Angelique R. Wise , Sara Maloney , Adam Hering , Sarah Zabala , Grace E. Richmond , Monica K VanKlompenberg , T. Murlidharan Nair and Jenifer R. Prosperi.

The work concerns a very important topic, namely chemotherapy for Patients with triple negative breast cancer (TNBC). As the authors point out, this is the first report in which Bcl-2 inhibition can restore sensitivity to paclitaxel in breast cancer cells with mutant Adenomatous Polyposis Coli. 

In my opinion, the article is very well written, organized, and the material is well selected. During a thorough review of the article, I found no errors.

The results are correctly presented and well visualized. The discussion was well conducted. The work is very interesting. A very important topic.

This research is critical to developing better treatment regimens for TNBC patients.

From a scientific point of view, I rate the work highly. I think that the work is suitable for printing in IJMS.

Author Response

Thank you for the kind review of our manuscript. 

Reviewer 2 Report

Comments and Suggestions for Authors

The manuscript "Bcl-2 up-regulation mediates taxane resistance downstream of APC loss" by Wise et. al, studies the role of Bcl-2 in mediating paclitaxel resistance in triple negative breast cancer cell lines with APC knock out. The authors find that upon APC knock out, these cells get resistant to paclitaxel induced apoptosis and that Bcl-2 upregulation is responsible for this drug resistance. Standard colony formation assays and wound healing assays are performed, which are standard in the field. The authors also use an inhibitor of Bcl-2, ABT-199 which can increase the apoptotic cells in combination with paclitaxel. Given the fact that most cases of breast cancer have APC mutations, the study is very significant in the field. The authors rightly knocked out APC in 2 TNBC cell lines and discovered these very important findings. The data is well presented with full blot pictures included in the supplemental file.  I would recommend accepting this manuscript for publication.

Author Response

Thank you for the kind comments on our manuscript. 

Reviewer 3 Report

Comments and Suggestions for Authors

The paper focuses on the role of APC in paclitaxel resistance and it highlights a possible combination with ABT-199.

The paper is suitable for publication in IJMS, I suggest some clarifications:

1. in Introduction, line 52, what do you mean with sporadic breast cancer? Can you quantify statistically the term "sporadic"?

2. in line 60 should you write "loss of APC correlates"?

3. a pictorial scheme as a synthesis of the interplay between targeted genes expression and their role in the listed cancers may improve readability;

4. check the caption of >Figure 2 for both typos and panels references;

5. your interesting result is presented in lines 165-167, are there possible side effects in a therapeutic scenario because of the involved molecular pathways with respect to the use of PTX or ABT-199 alone?

6. may this statement based on your previous study in line 212 depend on the specific tissue?

Author Response

Thank you for the comments and recommendations on our submitted manuscript. We’ve addressed the comments below:

  1. in Introduction, line 52, what do you mean with sporadic breast cancer? Can you quantify statistically the term "sporadic"?
    Sporadic breast cancer is any breast cancer that is not hereditary. This has been clarified in the document.
  2. in line 60 should you write "loss of APC correlates"?
    Thank you for noting this. I have made this change.
  3. a pictorial scheme as a synthesis of the interplay between targeted genes expression and their role in the listed cancers may improve readability;
    Because there are so few proteins involved here, we have chosen not to do this. I hope that is acceptable.
  4. check the caption of >Figure 2 for both typos and panels references;
    Thank you for noticing this. It looks like there was a formatting issue. This has been resolved.
  5. your interesting result is presented in lines 165-167, are there possible side effects in a therapeutic scenario because of the involved molecular pathways with respect to the use of PTX or ABT-199 alone?
    There definitely could be side effects with the drugs alone. Our goal is to be able to use lower doses of the drugs to avoid these side effects. Interrogation of side effects will be a focus of future in vivo studies.
  6. may this statement based on your previous study in line 212 depend on the specific tissue?
    That’s a great point – thank you. I’ve included that in the manuscript now as a possibility.